# High-Value Compounds in Fruit, Vegetable and Cereal Byproducts: An Overview of Potential Sustainable Reuse and Exploitation

**DOI:** 10.3390/molecules25132987

**Published:** 2020-06-30

**Authors:** Ali Zein Alabiden Tlais, Giuseppina Maria Fiorino, Andrea Polo, Pasquale Filannino, Raffaella Di Cagno

**Affiliations:** 1Faculty of Sciences and Technology, Libera Università di Bolzano, 39100 Bolzano, Italy; AliZeinAlabiden.Tlais@natec.unibz.it (A.Z.A.T.); GiuseppinaMaria.Fiorino@unibz.it (G.M.F.); Andrea.Polo@unibz.it (A.P.); 2Department of Soil, Plant and Food Science, University of Bari Aldo Moro, 70121 Bari, Italy; pasquale.filannino1@uniba.it

**Keywords:** food waste, high-value compounds, plant byproducts, fermentation, enzymatic treatments

## Abstract

Food waste (FW) represents a global and ever-growing issue that is attracting more attention due to its environmental, ethical, social and economic implications. Although a valuable quantity of bioactive components is still present in the residuals, nowadays most FW is destined for animal feeding, landfill disposal, composting and incineration. Aiming to valorize and recycle food byproducts, the development of novel and sustainable strategies to reduce the annual food loss appears an urgent need. In particular, plant byproducts are a plentiful source of high-value compounds that may be exploited as natural antioxidants, preservatives and supplements in the food industry, pharmaceuticals and cosmetics. In this review, a comprehensive overview of the main bioactive compounds in fruit, vegetable and cereal byproducts is provided. Additionally, the natural and suitable application of tailored enzymatic treatments and fermentation to recover high-value compounds from plant byproducts is discussed. Based on these promising strategies, a future expansion of green biotechnologies to revalorize the high quantity of byproducts is highly encouraging to reduce the food waste/losses and promote benefits on human health.

## 1. Introduction

Food waste (FW) is a global issue of increasing concern. The term refers to the organic waste produced during the harvest, processing and distribution of food products, and derived from domestic/commercial kitchens [1]. The Food and Agricultural Organization (FAO) of the United Nations defines food losses as: “the decrease in edible food mass throughout the part of the supply chain that specifically leads to edible food for human consumption” [2]. Indeed, it is estimated that ca. 45% of the food produced is lost or wasted before and after reaching the consumer, along all the steps of the food supply chains (primary production, post-harvest, processing distribution and consumption) [3]. 

In recent years, the issue has attracted even more attention due to both the large amount of FW produced and the alarming prediction of the growing trend of quantity will be produced in the next 25 years [1]. Every year, approximately 1.3 billion tons of FW, which is equal to one third of total food production worldwide, is discharged and/or lost, including fresh vegetables, fruits, meat, fish/seafood, bakery and dairy products [1,4]. Among the factors leading to the large amount of FW, the growing global population, accompanied by a higher food demand and a loss of food products throughout food supply chains, play a significant role.

The European Union (EU) provides legislative support aimed at preventing, reducing and managing FW [5]. Directive 2008/98/EC establishes a hierarchy for the appropriate management of waste, with the primary purpose of encouraging the prevention (Figure 1). According to an order of priority, in the framework of the waste hierarchy the governments should prefer the reduction of FW, and only as a second option subject it to preparation for re-use, recycling, recovery and disposal. Moreover, member states are requested to adopt three points for the appropriate management of bio and FW: (i) the separate collection of bio-waste with a view to the composting and digestion of bio-waste; (ii) the treatment of bio-waste in a way that fulfils a high level of environmental protection; and (iii) the use of environmentally safe materials produced from bio-waste [3,5]. 

However, nowadays the majority of FW is still being landfilled, composted and incinerated instead of recovered [3]. FW can be incinerated for the production of energy but the release of dioxins due to its high content of moisture and CO_2_ emission during the combustion may affect the human health and exert an environmental impact with dangerous effects [3,4]. Additionally, it has been observed that FW in the landfill can cause air pollution, bad smell and leaching as well as production of methane. Indeed, FW decomposes in less time compared to other organic waste present in the landfill, with a consequent higher methane generation [4]. Production of animal feed addressed to the breeding and of biofertilizers for agricultural practices are alternatives to repurpose and recycle FW. However, the use of FW for animal feeding is currently regulated in the EU, thus limiting its application in this field [6]. On the other side, the valorization of FW into biofertilizers, although suitable, should be improved in terms of production efficiency, with better process control and adoption of advanced technologies. Consequently, the development of novel and sustainable strategies to reduce the quantity of FW and improve its recycling is an urgent need. The high content of valuable components in food byproducts may represent a new reading key. Bioactive compounds (BC) and fibers may be extracted and recovered from food related byproducts and valorized into food ingredients, supplements and/or nutraceutical formulations, contributing to FW reduction and also providing an economic and nutritional benefits. In particular, fruit/vegetable and cereal by- products contain a significant amount of bio-components (e.g., proteins, polysaccharides, polyphenols and fatty acids) that can be exploited as antioxidants, additives and colorants for the formulation of novel and functional products [7,8].

Fruits, vegetables and cereals represent a small group in the plant kingdom that can be used for human consumption. Growing worldwide population and the recent dietary changes have led to a significant interest by humans in the consumption of fruits and vegetables while cereals and grains represent one of the most traditionally cultivated crops. Consequently, nowadays fruit, vegetable and cereal byproducts represent one of the main FW produced around the world. According to the most recent data provided by the FAO, the worldwide production of fruit, vegetable and cereal products has achieved surprisingly high values in the last decades, and food demand will continue to grow [9]. The great productions are accompanied by a parallel loss and waste occurring in all phases of the supply chain with a significant production of skin, seeds, peel, rind, pomace as fruit and vegetable byproducts, and husk, bran and germ as the main cereal byproducts [10]. 

In this review, a comprehensive overview on the chemical composition of high-value compounds from main fruit/vegetable and cereal wastes, considered as plant byproducts, is reported. Starting from this awareness, a potential exploitation of these components by fermentation and enzymatic treatments is discussed (Figure 2).

## 2. High-Value Compounds (HVC) in Plant Byproducts

In the last few decades, several studies on fruit, vegetable and cereal byproducts have suggested the presence in these substrata of a wide range of HVC, such as BC and dietary fibers, in different residual fractions [7,8]. Identification, quantification and extraction of these compounds, and the evaluation of their effects, have become major health- and medical-related research, revealing antioxidant, antimicrobial, anti-inflammatory, anti-immunomodulatory and anticancer properties [7]. HVC of plants are divided into two groups: essential (primary) and non-essential (secondary) metabolites. The former includes mainly vitamins and minerals that, in the human body, are capable of preventing deficiency diseases and maintaining specific biochemical processes. Phenolics, carotenoids, phytosterols, saponins, essential oils, and phytic acids are some of the main examples of secondary metabolites, which have important biological roles in achieving and maintaining optimal cellular health, leading to an improvement in longevity. Fruit, vegetable and cereal byproducts are mainly rich in phenolics, carotenoids, phytosterols, polysaccharides, proteins and fatty acids [7,10]. Table 1 summarizes quantitative data about BC and dietary fibers found in industrial fruit and vegetable byproducts and their antioxidant activity. Table 2 shows the main BC in cereal byproducts with known beneficial properties and current applications. In the following sections, different HVC in plant byproducts are reviewed.

### 2.1. Phenolic Compounds

Phenolic compounds are among the largest classes of phytochemicals with biological functions in humans. They have received considerable attention due to their capacity to replace synthetic preservatives because of their ability to scavenge free radicals and prevent oxidation reactions in food. From a structural point of view, they all have in common at least one aromatic ring bearing one or more hydroxyl groups. The concentration of polyphenols is often higher in fruit and vegetable wastes compared to their edible parts. Polyphenols have high capacity to prevent cell damage and oxidative stress. Phenolic acids, flavonoids, and tannins are the most widely distributed phenolic compounds available in fruit and vegetable byproducts [8]. 

Phenolic acids, also known as phenol carboxylic acids, are aromatic acids that contain a phenolic ring and an organic carboxylic acid (C6–C1 skeleton). They can be subdivided into two families: hydroxybenzoic acids (C6-C1) and hydroxycinnamic acids (C6-C3). Hydroxycinnamic acids are generally more abundant in fruits and vegetables when compared to hydroxybenzoic acids [8]. Caffeic, ferulic, *p*-coumaric and sinapic acids are the most common representatives of hydroxycinnamic acids. Hydroxybenzoic acid derivatives are mainly present in the form of glucosides, such as *p*-hydroxybenzoic acid, vanilic and protocatechuic acid [40]. Among the variety of phenolic compounds, phenolic acids have attracted considerable interest due to their health benefits towards chronic degenerative diseases, and for their impact on multiple sensory (e.g., flavor, astringency, color) attributes of plant-based foods and beverages [7,8]. 

Flavonoids have the general structure of a 15-carbon skeleton, which consists of two aromatic rings (A and B), linked through a three-carbon chain (C), usually in the form of a heterocyclic ring. Different substitution patterns in ring C create various subclasses that are present in fruits and/or vegetables, like flavonols, flavones, flavanones, isoflavones, flavanonols, and anthocyanins [8]. Flavonoids are capable to form complexes with both extracellular and soluble proteins as well as bacterial cell walls, which display high antimicrobial activity. However, they exhibit high antioxidant activity because of their redox potential, which allows them to act as reducing agents and hydrogen quenchers [40]. 

Tannins are water-soluble phenolics that have a tendency to interact with aqueous solutions of proteins to form insoluble precipitates [8]. Tannins can be subdivided into four major groups based on their structural features: gallotannins, ellagitannins, complex tannins and condensed tannins. Epicatechin and catechin are the most widely studied condensed tannins [8]. Despite antinutritional effects, tannins compounds have a great role as antioxidant due to the multiple phenolic hydroxyl groups, able to reduce the concentration of free radicals. In addition, they may exert antimicrobial activity through the inhibition of enzyme activities, depletion of metal ions and precipitation of membrane proteins [41]. 

### 2.2. Carotenoids

Carotenoids are lipid soluble pigments, biosynthesized and localized in the chloroplasts and chromoplasts of plants, including fruits and vegetables. Some carotenoids are used as natural pigments during food processing. Carotenoids share a similar base structure, which is composed of eight isoprene units and characterized by a series of conjugated double bonds. Two types of carotenoids exist, which differ only in their oxygen content: xanthophylls and carotenes [8]. Lutein, which is usually found in dark green leafy vegetables, is one of the most common xanthophylls [8]. About carotenes, orange carrots, orange sweet potato, nectarines and mangoes are the most well-known dietary sources of β-carotene, while watermelon, tomatoes, pink grapefruit and red papaya are some of the greatest sources of lycopene. High concentrations of carotenoids derived from these fruit and vegetable processing wastes find applications in pharmaceutical uses. In carotenoids, the conjugated double bond arrangement is primarily responsible for their chemical reactivity with free radicals enabling to suppress their damage. Moreover, one research group [42] has reported that some carotenoids are able to reduce the risk of certain cancers, potentiate the functions of the immune system, protect against cataract formation, and prevent vitamin A deficiency. 

### 2.3. Phytosterols

Phytosterols is a collective term to indicate BC present exclusively in plants. From a chemical point of view, phytosterols belong to the triterpene family and their chemical structure is composed of a tetracyclic structure and a side chain in position C17 [43]. Phytosterols can be sub-divided into two classes, plant sterols or “Δ5-sterols” with a double bond at the 5 position in the sterol ring, and plant stanols or 5α-sterols with a saturated sterol ring structure. They are structurally similar to cholesterol, and are widely accepted to be the most abundant sterols in human cells, although plant cells also contain it in various concentrations and forms [43]. More than 200 different types of phytosterols have been identified in plants. β-sitosterol (24-α-ethylcholesterol), campesterol (24-α-methylcholesterol) and stigmasterol (Δ22, 24-α-ethylcholesterol) are considered the most abundant plant sterols, even if in plants the phytosterols content and their conjugated forms can significantly depend of environmental factors [43]. Relevant amounts of these BC have also been found in seeds, legumes, vegetables and unrefined vegetable oils. Oilseed and dried fruit showed a high content of phytosterols. Some authors reported a very high concentration of phytosterols in pistachio oil (271.9 mg 100 g^−1^ of oil) while other evidence showed a lower concentration of total phytosterols in walnut and almond (ca. 140–160 mg 100 g^−1^ of oil) [44,45]. Considerable amounts of phytosterols have also been found in food byproducts such as rice and wheat brans, wheat germ, and oat hull. The beneficial effects of phytosterols on human health have been widely observed. Since the 1950s hypercholesterolemic individuals have been treated using drugs and/or supplements containing β-sitosterol with a promising reduction of serum cholesterol levels. Nowadays, in addition to the hypocholesterolemic role, it has been reported that plant sterols can be used for the treatment of prostatic diseases, for anti-inflammatory purposes, for inhibition of cancer, and for anti-oxidative effects [46].

### 2.4. Phytic Acid

Phytic acid is found in several plant byproducts, including cereals, legumes, vegetables and nuts [47]. From a chemical point of view, it is composed of six phosphate groups linked to an inositol ring. When all six carbons are bonded to phosphate groups, the compound becomes known as inositol hexaphosphate, while it is called phytate if it is in salt form [47]. Phytic acid was recently included among BC. For a long time, phytic acid was considered to be an antinutrient due to its capability to interact and bind to essential minerals, thus limiting their absorption in the intestinal tract [47]. Nevertheless, recently, attention has been focused on its positive health effects. In fact, it is a natural antioxidant agent, and it provides numerous protective effects toward several diseases, including the reduction of serum lipids and inflammatory processes [47]. In addition, the application of phytic acid in foods is increasing more and more due to its ability as natural preservative. 

### 2.5. Dietary Fibers

The Codex Alimentarius by FAO and WHO classifies dietary fibers as non-starch polysaccharides with 10 or more monomeric units that are resistant to digestion and absorption in the human small intestine, and with complete or partial fermentation in the large intestine [48]. Dietary fibers provide structural rigidity to the plant due to their location in the plant cell wall. Based on chemical, physical and functional properties, they can be classified into two groups: water-soluble (e.g., pectin) and water-insoluble (e.g., cellulose and lignin). The ratio of insoluble/soluble polysaccharides and their polymeric composition vary depending on the class and origin of the plant, developmental stage and environmental conditions during growth. Byproducts from fruit processing are composed of cellulose (40–50%), hemicellulose (20–30%) and lignin (10–25%), along with other polysaccharides. Cellulose is a linear polysaccharide composed of β-(1→4)-linked glucose monomers with individual chain aggregation to form micro fibrils via hydrogen bonding. Due to their sugar lowering effects, antibody immobilization, water retention and biodegradability, cellulose and its derivatives have been used in diagnostic and biomedical applications. Hemicelluloses, the second most abundant biopolymer, cross-link cellulose microfibrils via hydrogen bonding resulting in the formation of xyloglucans, xylans, mannas and glucomannans [49]. Oligosaccharides such as xylooligosaccharides, which are produced upon controlled hydrolysis of xylan, are resistant to action by human gut enzymes and pass undigested into the colon, wherein they are fermented by colonic microbiota. Fermentation of such oligosaccharides yields mainly short-chain fatty acids (SCFAs) (acetate, propionate, and butyrate), which have beneficial effects on the intestinal epithelium and gut immune system [49]. β-glucans are bioactive polysaccharides found mainly in cereals, but also in fruits and legumes [27]. They are linear polymers of glucose units linked, in cereals, by β 1-4 and β 1-3 glycosidic bonds. Based on molecular mass, viscosity and solubility, they present various differences, thus exerting several physiological effects in animals. Higher amounts of β-glucans were found in barley (3–11%) and oat (3–7%) compared to rye (1–2%) and wheat (<1%). It was observed that the high viscosity of β-glucans prevents the absorption of lipids or the reabsorption of bile acids and their metabolites with a consequent reduction of the cholesterol serum levels. Nutrition guidelines from United States (Dietary Guidelines for Americans) and Europe (European Food Safety Authority Scientific Opinion) strongly recommend a daily intake of 25–30 g of dietary fibers, but the most commonly consumed foods are characterized by low concentrations [50]. In this context, the addition of pomace byproducts with fiber contents to food formulations should be highly considered to develop functional foods with additional health benefits. 

## 3. Main Plant Byproducts

Most food wastes are generated by industrial processing—so-called byproducts—and might be a good source for high added-value bioactive compounds. Fruit and vegetable industrial wastes are mainly constituted of peels, pomace, seed fractions, and rind; whereas husk, bran and germ are the main cereal industrial wastes [7]. In the following section, the main fruit, vegetable and cereal byproducts are described, highlighting their sources, bioactive compound contents, and biological and functional properties.

### 3.1. Apple Byproducts

Apples (*Malus domestica* sp.) are one of the most important fruit crops worldwide. The total world apple production for 2017 was about 83.1 million tons [7]. An important part of this production is destined to become juice and derivative products, such as cider, jams, vinegar and dried fruits. Consequently, large quantities of apple pomace are generated each year as agro-industrial byproducts as a result of the production process, which need to be disposed of [12,13]. Apple byproducts contain mainly peels/flesh (95%), seeds (2–4%), and stem (1%) [11]. In recent years, research efforts have been devoted to studying the composition and properties of apple pomace, revealing interesting contents in terms of nutrients, phytochemicals and functional components, in addition to being an important source of pectins and fiber [13]. Apple byproducts contain high quantities of fibers (mainly insoluble fibers like cellulose, 7–40%, hemicellulose, 4–25%, and lignins, 15–25%) and substantial amounts of pectin (5–10%). Moreover, apple byproducts are an important source of bioavailable polyphenols, mainly flavanols (monomeric and oligomeric), dihydrochalcones and anthocyanidins [12]. The polyphenolic content differs greatly between crop varieties and different parts of the apple; for instance, apple flesh contains a lower concentration than peels. The most abundant polyphenols in apple byproducts are chlorogenic acid, phloretin glucosides and quercetin glucosides. Other polyphenolic compounds such as catechins and procyanidins have been found in low amounts [51]. The BC in apple byproducts is potentially able to affect human digestion and metabolism, cholesterol and triglyceride homeostasis, and diabetes and sex hormone production. Moreover, they have antioxidant, anti-inflammatory, antiproliferative, antibacterial and antiviral effects [7,51]. 

### 3.2. Avocado Byproducts

Avocado fruit (*Persea americana* Mill.) is a crop indigenous to tropical and Mediterranean regions. Because of the easy cultivation and management, agreeable flavor and high nutritional value, avocado has gained broad recognition as an oleaginous fruit crop, taking on great importance in the human diet. Industrially, avocados are processed into guacamole, paste, and oil, generating large quantities of agro-industrial byproducts, mainly peel and seed, ranging from 18% to 23% of the original fruit dry weight [14]. Avocado peel and seed represent an excellent natural source of bioactive phytochemicals such as phenolic acids and flavonoids, including procyanidins, flavonols, hydroxybenzoic and hydroxycinnamic acids [14]. Peel antioxidant activity exhibits higher values than the seeds and pulp due to its threefold increased polyphenol content [7]. On the other hand, a lipid extract of the avocado seed exhibited highly powerful anti-inflammatory and anti-cancer effects due to the high percentages of hydrocarbon, sterols and unsaturated fatty acids, mainly oleic acid [52].

### 3.3. Banana Byproducts

Banana (*Musa* spp.) is one of the most produced and consumed fruits worldwide. Recently, the FAO reported that global banana production reached 114 million tons in 2017 [7]. High quantities of the produced crops are industrially processed, thus generating byproducts like peels, rhizomes, fruit stalks, inflorescences, leaves, and pseudo-stems [7]. For instance, banana peel is a household and industrial FW that is discarded in large quantities, accounting for 30–40% of the total weight of the ripe fruit. Most of these byproducts have macromolecules that are valuable for the food and pharmaceutical industries [7,15]. The banana peel is rich in dietary fibers (cellulose, lignin, resistant starch, pectin, hemicelluloses), protein, essential amino acids, polyunsaturated fatty acids and potassium [15]. Moreover, it is a natural source of antioxidant compounds including hydroxycinnamic acids, flavonoids, phytosterols, carotenoids, anthocyanins, and vitamins. These compounds can have a role against aging and various degenerative diseases [7].

### 3.4. Pomegranate Byproducts

Pomegranate (*Punica granatum* L.) is an ancient, mystical, and unique fruit that is native to the Middle East. Pomegranates are commonly consumed in the form of juices, jams, jelly squash, syrup, and nectar. Pomegranate peels are the main byproduct obtained through juice processing, together with pulp and bagasses [53]. Recent studies on the functional and nutraceutical properties of pomegranate peel suggest a significant presence of ellagitannins, flavonoids and anthocyanins, which have important health significance [53]. Ellagitannins, the main hydrolysable tannins of pomegranate, possess antimicrobial, antioxidant, anti-mutagenic, anticancer, anti-inflammatory, anti-diabetic, and other properties beneficial to health [16]. 

### 3.5. Date Byproducts

Date palm (*Phoenix dactylifera* L.) is one of the most extensively cultivated crops in arid and semi-arid regions around the world. This fruit plays an important role in human nutrition, since it provides a wide range of essential nutrients and represents a very good source of carbohydrates, dietary fibers, vitamins, fatty acids and proteins. Date palm is composed of flesh (pulp) and seed (pits). Date seed represents 10-15% of the total fruit weight, depending on maturity, variety and grade [17]. As the main and general waste byproduct of dates, the seed is an excellent source of dietary fibers (characterized by a high level of water-insoluble mannan fibers) and contains considerable amounts of protein and lipids. A plethora of biological activities have been attributed to the natural phenolic and antioxidant compounds contained in date seeds, mainly related to protection against illnesses such as cancer and cardiovascular disease [17].

### 3.6. Mango Byproducts

Mango (*Mangifera indica* L.), one of the most popular exotic fruits, is cultivated in many parts of the world, particularly in tropical countries. Each part of the mango fruit contains essential nutrients that can be used with positive effects on human health. Puree, slices in syrup, nectar, pickles, and canned slices are the main industrial products obtained from mango fruits, while the major waste byproducts are the peel and seeds [7,20]. The peel represents 7–24% of the total weight of the mango fruit. Recently, mango peel has aroused interest among the scientific community due to its high content of valuable compounds, such as dietary fibers, carotenoids and vitamins, which have prominent functional and antioxidant properties [7]. Moreover, mango peel is a good source of polyphenols such as hydrolysable tannins, flavonoids, proanthocyanidins and alkylresorcinols [19]. On the other hand, the seed accounts for approximately 20% of the whole fruit, and it is a safe and natural source of pectin, edible fats, proteins, polyphenols and fatty acids like stearic and oleic acid [18]. Anticancer activity against breast and colon cancer and antimicrobial activity against Gram-positive and Gram-negative bacteria have been reported for the polyphenols contained in mango seed (in particular mangiferin, quercetin, kaempferol, anthocyanins, ellagic, gallic, protocatechuic and ferulic acids) [18].

### 3.7. Citrus Byproducts

Citrus fruits (*Citrus* spp.) are usually processed to produce juice, generating a huge amount of byproducts, mainly peel and seeds. Peel can account for about 60–65% of the total weight of citrus byproducts, with internal tissues amounting to 30–35% and seeds accounting for 0–10% [7]. BC such as ascorbic acid, natural flavonoids (hesperidin, naringin and narirutin), flavonols (rutin and quercetin) and flavones (diosmin and tangeretin) with antioxidant properties and great importance for human nutrition have been found in citrus byproducts [21]. Moreover, citrus peel is a potential source of dietary fiber and polysaccharides, with water and oil-holding capacity as well as swelling properties that are much higher than those of cellulose. Over the last decade, citrus peel waste has been reported to be an excellent source of essential oils, with D-limonene being the major component (32–98% of total oils), along with natural antioxidants, ethanol, organic acids, pectic oligosaccharides and pectin [7,22].

### 3.8. Tomato Byproducts

Tomato (*Solanum lycopersicum*) is an annual crop that is widely cultivated in temperate climates around the world. Being one of the most consumed vegetables, both fresh and in the form of processed food products, huge amounts of tomatoes are processed annually to produce juice, sauces, purees, pastes, and canned tomatoes, resulting in large quantities of solid wastes such as vascular tissues, seeds and peels [7,23]. These byproducts, known as tomato pomace, are a promising source of compounds that could be used for their nutritional properties and antioxidant potential, such as carbohydrates, organic acids, pigments, fiber, proteins, oils and vitamins [7]. Furthermore, tomato byproducts are a great source of lycopene, which has demonstrated antioxidant properties and potential role in the prevention of chronic diseases. In addition to lycopene, tomato peels also contain phenolic compounds with important biological activities, like caffeic, ferulic and chlorogenic acids, quercetin and quercetin-3-β-*O*-glycoside [23].

### 3.9. Carrot Byproducts

Carrot (*Daucus carota* L.) is recognized as an important root vegetable and is used commonly for juice production. Large amounts of carrot pomace (ca. 30–40% of weight) are generated during juice production. Carrot peel, the main constituent of carrot pomace, is known to be rich in phytonutrients, mainly carotenoids, polyphenols, vitamins, and minerals, which exhibit health promoting effects [24]. Carrot peel contains high amounts of dietary fibers, particularly pectin, a soluble dietary fiber consisting of d-galacturonic acid units, l-rhamnose, l-arabinose, and d-galactose, along with other monosaccharides [7]. In addition, contributions to human health have been attributed to BC from carrots thanks to their antioxidant, antimicrobial and antimutagenic activities [7,24].

### 3.10. Cauliflower Byproducts

Cauliflower (*Brassica oleracea* L. ssp. *Botrytis*) is a popular vegetable crop and that has an increasing production due to its nutritional and functional properties. Cauliflower has a very high waste index, and tons of cauliflower byproducts (stems and leaves) are generated after harvesting every year [54]. Cauliflower byproducts are well known to contain various beneficial molecules, such as dietary fiber, phenolic compounds, vitamin C, glucosinolates, carotenoids and leaf protein [49]. The major phenolic compounds in cauliflower byproducts are flavonoids (kaempferol and quercetin) and hydroxycinnamic (caffeic and sinapic) acids [7]. Several authors have studied numerous protein hydrolysates from cauliflower byproducts displaying antioxidant property, angiotensin I-converting enzyme inhibitory activities in cell-free systems, and an important role in the regulation of glucose consumption and glycogen content in HepG2 cells [54].

### 3.11. Wheat Byproducts

In recent years, it has been estimated that 12.9% of all food wastes are produced during the processing and manufacturing of cereals. Along the cereal food chain, ca. 10–12% of total production is lost in North America and Europe, while total loss and waste amounts to up to 18% in industrialized Asia. In general, it is estimated that the amount of cereal waste is ca. 40,000/45,000 tons/year in Europe [55]. Wheat belongs to the genus *Triticum* and is classified into either hard (*Triticum durum*) or soft (*Triticum aestivum*) varieties. Traditionally considered the most popular cereal for human consumption, its production accounts for ca. 29% of total cereal production [10]. Wheat bran is the main wheat byproduct obtained from milling and flour production [10]. Although the endosperm destined for flour production represents around 80% of the grain dry weight, the bran obtained as byproduct is ca. 15% [10]. It is known that wheat bran can provide gastrointestinal benefits, reduce the risk of metabolic disorders and cardiovascular diseases, and provide anticancer effects [56]. Its valuable phytochemical composition includes water (12%), proteins (13–18%), fats (3.5–3.9%) and carbohydrates (56%), with a significant amount of health-promoting bio-components such as dietary fibers, polyphenolic compounds, sterols, β-glucans, lignans and phytic acid [10]. Four main groups of compounds have been isolated from wheat bran: (i) soluble and insoluble dietary fibers, including arabinoxylan and β-glucans; (ii) sugars and their derivatives, including glucose, starch and succinic acid; (iii) secondary plant metabolites, including phenolic acids; and (iv) proteins [57]. Phenolic acids are contained in the bran layers, and they can be present as soluble free acids or soluble/insoluble conjugated moieties esterified into sugars/cell wall structural components [10]. The most abundant phenolics are benzoic acid derivatives (e.g., gallic, synergic and vanillic acids) and cinnamic acid derivatives (e.g., ferulic, *p*-coumaric, and caffeic acid), with a neat predominance of ferulic acid which accounts for 70–90% of the phenolic compound derivatives [10]. Another highly valuable source of BC is the wheat germ that accounts for ca. 2% of weight after wheat grain milling. Classified as a byproduct of the dry milling industry, it has attracted scientific attention due to its high content of beneficial molecules. Its bioactive chemical composition includes dietary fibers, amino acids, tocopherols, unsaturated fatty acids, essential fatty acids (linolenic acid) and polyphenolic compounds [10]. In addition, several studies have demonstrated that the use of wheat germ in food items (in particular bread) confers a high degree of added value and functional properties [58].

### 3.12. Rice Byproducts 

Rice (*Oryza sativa* L.) is the staple food for almost half of the world’s population and is cultivated on all continents. It represents an excellent source of BC, such as vitamins, minerals and phenolic acids, and flavonoid compounds, such as anthocyanins, γ-oryzanol, tocopherols, tocotrienols, sterols, and dietary-antioxidant fibers [10]. The rough rice processing that is carried out in order to obtain milled edible rice (generally known as white rice) generates byproducts including rice hull (20% of rough rice) and rice bran and germ (10% of rough rice) [59]. Currently, most produced rice hull (or husk) is not used for any other applications due to its high content of silicon and rapid decomposition by bacteria [10]. However, rice bran is a very rich source of BC, including γ-oryzanol, ferulic acid, phytosterols, tocols, γ-aminobutyric acid (GABA), and phytic acid. These biocomponents have shown cardiometabolic protection properties [60]. Finally, rice germ presents a high content of proteins, lipids and tocopherols.

### 3.13. Corn Byproducts 

Corn (*Zea mays*) represents a fundamental food for more than a billion people around the world, in particular for the populations of Latin America and Southern Africa [10]. From a nutritional point of view, corn can be considered an excellent source of thiamine (vitamin B1), pantothenic acid (vitamin B5), vitamin C, folate, phosphorus and manganese. In addition, its consumption can provide several BC, such as carotenoids, phenolic acids, dietary fibers, phytosterols and phytostanols. Corn is subjected to dry and wet milling processes in order to obtain the final product for food consumption. Both processes can lead to corn byproducts that are rich in BC. The main byproducts of the corn dry milling process are corn germ and corn bran (the latter reaching 60–70 g Kg^−1^ of corn byproducts) [10]. In corn bran, the main components are dietary fibers (e.g., heteroxylans) and phenolic acids, but other compounds such as starch (9–23% of total weight), proteins (10–13% of total weight), lipids (2-3% of total weight), and ash (2% of total weight) have also been found [10]. The concentration and type of phenolic acids in corn bran might change depending on the type of corn; in general, the main phenolic acids found in corn bran are ferulic acid, along with vanillic, caffeic, *p*-coumaric, and *p*-hydroxybenzoic acids [10]. Among flavonoids, anthocyanins have been found in abundance in corn bran, and their capability to provide a large spectrum of colorations can be exploited as food pigments. For instance, a recent study reported that pigments, including anthocyanins, extracted from purple corn pericarp can be successfully used in acidic foods and beverages with an acceptable shelf-life [61]. 

### 3.14. Barley Byproducts 

Barley (*Hordeum vulgare* L.) is considered to be one of the oldest cultivated cereal grains and it has, since ancient times, been grown to provide staple foods for human nutrition. In the modern world, the consumption of barley-related foods has undergone a decrease, except for its use in breweries and in the alcohol industry, where it represents a primary raw material. Recently, barley has attracted more scientific attention as it contains many BC with beneficial effects on human health, such as dietary fibers (especially β-glucans), tocols (including tocopherols and tocotrienols), and phenolic compounds [62]. According to the European Food Safety Authority (EFSA), the consumption of β-glucans from barley contributes to reducing the post-prandial serum glucose levels and to maintaining normal blood concentrations of LDL-cholesterol [63]. Barley also presents some unique phytochemical properties such as the presence of all tocols belonging to vitamin E family (i.e., α-, β-, γ- and δ-tocopherol and α-, β-, γ- and δ-tocotrienol), which are usually not found together in other cereals. Several studies have highlighted that barley grain contains different proportions of all four tocol vitamers, displaying antioxidant and anti-cancer properties [62]. For instance, pearling byproducts have shown tocopherols to be preferentially localized in the germ, whereas tocotrienols are concentrated in the other parts of the kernel (i.e., aleurone and sub-aleurone layer). Barley grain is also rich in phenolic compounds, in both free and bound forms. Catechin, procyanidins and prodelphinidins are the main representative compounds in the free phenolic fraction, while ferulic, *p*-coumaric and vanillic acids are major constituents of the bound phenolic fraction [62]. It is well known that these polyphenols have anti-inflammatory and vasodilating effects that improve the blood lipid profile, and that they reduce the antioxidant processes at the expense of low-density lipoproteins [62]. Barley bran also represents a plentiful source of dietary fibers, including β-glucans that can be used in food, as well as in cosmetic and pharmaceutical applications [64]. In recent years, particular attention has been paid to the recovery and re-use of brewer’s spent grain derived from beer production. Although its chemical composition can vary depending on barley variety, harvest time, malting and mashing conditions, it is generally composed of a lignocellulosic material rich in protein and fibers. Brewery-spent grain contains hemicellulose (21.8–28.4% of dry weight, mainly arabinoxylans), cellulose (16.8–25.4% of dry weight), lignin (11.9–27.8% of dry weight), protein (ca. 15.2–24.0% of dry weight) and a few percent of residual starch and starch-derived products (glucose and maltosaccharides) [65]. Brewery-spent grain also represents a good source of minerals (e.g., calcium, iron, magnesium, manganese, phosphorus, potassium, and sodium), vitamins (e.g., biotin, choline, folic acid, niacin, pantothenic acid, riboflavin) and amino acids (e.g., leucine, valine, alanine, serine, glycine, glutamic acid, and aspartic acid). Based on chemical characteristics, brewer’s spent grain can be used not only as a cheap ingredient for animal feed, but also for food applications. Due to its granular structure, brewer’s spent grain is firstly converted into flour and subsequently used as an additional ingredient in the manufacture of bakery products, resulting in an enrichment of fibers and proteins. Brewery-spent grain can be also incorporated in order to increase the fiber content of other food products (e.g., frankfurters). Finally, thanks to its carbohydrate composition, it has also been exploited as a suitable and alternative substrate for the production of various enzymes [66].

## 4. Exploitation of Plant Byproducts through Enzymatic Treatment and Fermentation

Based on the above-described nutritional features of plant byproducts, and considering their constituent volumes resulting from food processing, there is a clear need to dispose of them through sustainable means. Due to their high water content and important organic load, plant byproducts are prone to microbial decomposition, causing environmental issues when disposed of, and thus leading to additional waste treatment costs for food manufacturers [8]. The generation of valuable products from plant byproducts by means of biological processes, using microbes and enzymes, is a versatile, sustainable and promising route for counteracting the growing rate at which they need to be disposed of. In fact, biotechnological processes involving microbial fermentation and enzymatic treatment might mediate the chemical conversion of byproducts, producing industrially important compounds with high biological activities. Biotechnological processing has largely used byproducts as substrates for enzyme-catalyzed production of different chemicals such as citric acid, biofuel, flavor components and pigments. The incorporation of food byproducts as natural preservatives and nutraceuticals in various food preparations has also been widely reported in the literature. Recent studies have also revealed the direct extraction and characterization of antioxidant biomolecules and BC obtained through non-fermentative processes [7,8]. The recovery of biogenic compounds from low-cost plant byproducts may generate economic and environmental benefits, and companies might receive a financial return, instead of incurring the costs of disposal. A graphical illustration of the fate of fruit, vegetable and cereal wastes after enzymatic treatment and fermentation is shown in Figure 2. In the following sections, some examples of plant byproduct exploitation through these biotechnological processes are discussed.

### 4.1. Enzymatic Treatment

One of the greatest challenges is the development of effective methods for BC extraction and their release from the plant matrix. Indeed, plant cell walls can reduce the extraction efficiency and the application of optimized, and comprehensive protocols for the enhanced recovery of bioactive substances appears to be an essential prerequisite [67]. In this regard, enzyme-based extraction is a potential alternative that makes it possible to overcome several drawbacks of conventional solvent-based methods (e.g., low extraction yields; long extraction times; presence of organic solvents in the products). Enzymes represent a suitable group of catalysts derived from bacteria, fungi, animal organs or vegetable/fruit extracts, which can be involved in the extraction, modification or synthesis of natural BC. Various enzymes such as cellulases, pectinases and hemicellulases are able to disrupt and degrade the structural integrity of the plant cell wall, thus enabling better release and efficient extraction of bioactives [67]. Moreover, enzymes can catalyze reactions with high specificity and regioselectivity under mild conditions, such as in aqueous solution. Enzymes offer the possibility of greener chemistry, with pressure mounting on the food industry and even on pharmaceutical companies to identify cleaner routes for the extraction of new compounds [58]. In the last two decades, many studies have investigated the possibility of using several classes of enzymes for FW treatment for the recovery of major bioactive components from fruit, vegetable and cereal byproducts (Table 3). Commercial cellulases and pectinolytic formulations (e.g., Viscozyme L cellulolytic enzyme complex, Ultrazym pectinolytic preparation, Pectinex and Lallzyme Beta), and/or microbial enzymes (e.g., α-amylase, β-glucosidase, xylanase, β-glucanase) have been successfully applied to improve the release of biologically active substances from food byproducts [68].

Among enzymes, proteases represent a group of hydrolases that are often used, both as commercial and microbial origins, for the production of bioactive peptides from fish and meat waste processing. Amylases are polysaccharide hydrolases that can be used to hydrolyze starch into sugars, which can be used for the production of various chemicals (e.g., ethanol) through fermentative processes. Moreover, with the rising importance of enzyme-assisted extraction of bioactives, researchers are now being attracted by the benefits of other non-conventional extraction techniques. Novel methods such as three-phase partitioning, microwave-assisted extraction, ultrasound-assisted extraction, supercritical fluid extraction, ionic liquid extraction, pulsed electric field (PEF), and high-pressure extraction have already been explored in the past to obtain high added-value products. The combination of enzymatic extraction with these green techniques can enhance these advantages. In the next section, a general overview of enzymatic treatment for the release of bioactive components from fruit, vegetable and cereal byproducts is discussed, focusing on the main BC previously mentioned.

#### 4.1.1. Enzymatic Treatments of Fruit/Vegetable Byproducts

Phenolic acids and flavonoids are important antioxidant compounds, and a considerable amount are concentrated in fruit and vegetable byproducts like peels, seeds, pomace and leaves. It is widely recognized that phenols can be recovered from the food matrix through enzymatic treatment. For instance, the application of enzymatic hydrolysis offers the chance to increase the extractability of polyphenols, including non-extractable proanthocyanidins, with the use of neither organic solvents nor other toxic chemicals. Three enzymes, such as pectinase, cellulase and tannase, were used to collect and characterize proanthocyanidins located in both the skins and seeds of grape berries, demonstrating that the addition of these enzymes to the extraction mixture increased the total content of phenols [69]. A higher recovery of polyphenols from grape seeds was achieved by the optimization of oenological enzyme preparations composed by cellulases and pectinases [70]. More recently, it was reported that in grape pomace, tannase enriches the phenolic extract in gallic and syringic acids while cellulase enriches it in *p*-coumaric acid and malvidin-3-*O*-glucoside [71]. Enzyme-assisted extraction through the single or combined use of pectinase, cellulase, tannase and β-glucosidase has been successfully applied for the recovery of phenols from citrus byproducts [72]. In addition to the recovery of polyphenols, bergamot peel has been subjected to enzymatic treatment by using three fungal enzyme preparations from *Aspergillus* sp. and *Trichoderma* sp. (i.e., Pectinase 62L, Pectinase 690L and Cellulase CO13) for the release of low molecular weight pectic oligosaccharides and a mixture of glycosylated and deglycosylated flavonoids [73]. Enzyme-assisted supercritical fluid extraction is also an efficient and eco-friendly green process for the optimal extraction and recovery of phenolics and other antioxidants from pomegranate peels. The use of enzymes was also efficient for recovering BC from the leaves of vegetables. For instance, two commercially available polysaccharide-degrading enzymes, Viscozyme L and Rapidase, were used for the release of phenolics from the outer leaves of cauliflower, resulting in a significantly higher extraction yield of kaempferol-glucosides as compared to the control treatment [75]. The use of enzymes has also been applied for the recovery of carotenoids, polysaccharides and dietary fibers from fruit/vegetable byproducts. Cellulase and pectinase have been widely employed as a pretreatment step of tomato-based products prior to solvent extraction for the recovery of carotenoids, especially lycopene. Pretreatment of tomato waste using a food-grade enzyme preparation with pectinolytic, cellulolytic and hemicellulotic activities resulted in an 8- to 18-fold increase in the recovery of lycopene in extraction yields [68]. Enzymatic catalysis can also be applied to obtain pectin products with desirable functional properties. Several studies have suggested that enzymatic treatment can substitute the traditional acid-based extraction in terms of pectin release from the peel. For instance, six commercial cellulases were screened to recover pectin from lime peel biomass; the screening demonstrated that the most efficient enzyme preparation was Laminex C2K derived from *Penicillium funiculosum*, which released pectin with a similar yield and functional properties as the classically acid-extracted pectins [74]. Recently, another commercial multicatalytic enzyme preparation, named Celluclast 1.5 L, was successfully used to collect pectin from apple pomace generated from the processing of apples, eliminating the requirement of a low pH and high extraction temperatures [76]. The enzymatic treatment based on Celluclast 1.5 L has also been adopted for pectin recovery in other byproducts [77]. In addition to the recovery of BC from vegetables and fruits, it has been observed that enzymatic pretreatment of FW exerts a positive effect also on lactic acid fermentation. Three commercial enzymes, Viscozyme, Flavourzyme and Palatase (containing a wide range of carbohydrases, a fungal protease/peptidase, and a purified bacterial lipase, respectively) have been used to treat vegetable and grain waste from cafeterias, resulting in a volatile fatty acid production higher than that of the control fermenter [84]. This result was explained as being a consequence of the rapid conversion of various soluble organics generated during the previous enzymatic pretreatment into volatile fatty acids. Enzymatic treatment based on amyloglucosidase and carbohydrase can be a useful approach for ethanol production from FW, since it contains a significant amount of soluble and insoluble sugars. High ethanol quantities were obtained from batch alcoholic fermentation by *Saccharomyces cerevisiae* of broth pretreated with the above enzyme mixture, suggesting that the use of FW might be more economical than using other biomasses for ethanol production [78]. Consequently, the conversion of FW into fermentable sugars represents a suitable method for bioethanol production, with FW being a promising substrate. For instance, highly fermentable sugars have been produced from FW by using enzymatic hydrolysis with glucoamylase and a prior hydrothermal and acidic treatment (with diluted hydrochloric and sulphuric acids) that degraded larger molecules of polysaccharide.

#### 4.1.2. Enzymatic Treatment of Cereal Byproducts

Rice bran has been recognized as an excellent source of BC with relevant antioxidant activity, but its rough texture limits its use [79]. Several BC located in the bran layer, such as oryzanols, tocopherols, tocotrienols, and phenolic compounds with antioxidant properties, could be recovered and exploited to develop nutraceuticals and functional foods. Germination coupled with enzymatic treatment of rice bran using Protease type II from *Aspergillus oryzae* and α-amylase type XII-A from *Bacillus licheniformis* can considerably improve the concentrations of γ-oryzanol, α-tocopherol and phenolics compared with those of untreated rice bran, with a consequent increase of the antioxidant value [79]. Recently, three different treatments, namely with hot air, far-infrared radiation, and cellulase, have been applied in order to investigate the changes of antioxidant activity and BC in rice bran, rice husk and ground rice husk [80]. Although far-infrared radiation was shown to be the most promising method, cellulase significantly increased the amount of vanillic acid and γ-oryzanol. Rice bran also represents a good source of dietary fibers, mainly cellulose, hemicellulose, lignin and pectic substances, which provide beneficial effects, maintaining gastrointestinal function, lowering the postprandial glycemic index and reducing the risks of cardiovascular diseases, diverticulosis and colon cancer. The addition of these dietary fibers to recipes could functionalize food products. For instance, when rice bran and oat bran were treated with different levels (0.70 and 700 ppm) of an endoxylanase enzyme and subsequently incorporated into cakes, this resultin a neat valorization of their nutritional value and technological quality characteristics (i.e., an increase in batter viscosity and gelatinization temperature, a decrease in water activity, and an improvement in volume, porosity, texture and sensory characteristics). In addition to rice bran, wheat bran also represents a reusable source of dietary fibers (xylans, lignin, cellulose, and galactans, fructans), vitamins, minerals and several BC (e.g., alkylresorcinols, ferulic acid, flavonoids, carotenoids, lignans and sterols). In recent years, it has been found that the application of enzymes produced by *Trichoderma* species can be successfully adopted to increase the content of soluble dietary fibers in cereal products. Indeed, *Trichoderma* species are efficient producers of extra-cellular enzymes, generating chitinases, glucanases, xylanases and cellulases that are able to hydrolyze a variety of plant materials. Cereal products with high contents of dietary fiber have been successfully treated with such enzymes, obtaining a significant conversion of insoluble dietary fibers into soluble ones. Moreover, it was also possible to observe a release of hydroxycinnamic acids, mainly ferulic acid strictly linked to the polysaccharide chains, with concomitantly increasing water-soluble antioxidant activity and phenol compound bioavailability. Enzyme preparation from thermophilic fungus *Humicola insolens* (namely Ultraflo L) efficiently solubilized brewer’s grain and wheat bran with a release of ferulic acid (almost all in the free form), *p*-coumaric acid (accounting for 50% in wheat bran and 9% in brewer’s grain), cellulose and arabinoxylan [81]. Brewery-spent grain can be also subjected to enzymatic treatment using carbohydrates and peptidase, leading to the solubilization of carbohydrate- and protein-derived fractions [82]. This outcome opens up promising applications for brewer’s spent grain derivatives in the food industry, for instance, as a prebiotic food supplement (i.e., the carbohydrate fraction) or as a food or feed supplement (i.e., the peptide fraction). Cereal byproducts can also be treated with enzymatic preparations in order to hydrolyze polysaccharides and yield sugar feedstock that can be transformed (through fermentation) into ethanol. Enzymes with amylolytic and hemicellulolytic/cellulolytic activities can hydrolyze carbohydrates, as they are cereals rich in such components. In wheat and rye bran, complex enzyme systems containing amylase (e.g., Vilzim SKA), glucoamylase (e.g., Vilzim SKG) and xylanase preparations with cellulase β-glucanase activities (e.g., Vilzim SKK) led to an increase in ethanol content and to a reduction in methanol concentration and fuel oil content (propanol, isobutanol, isoamyl and amyl alcohols) [83].

### 4.2. Microbial Fermentation

The application of microbes relies on their ability to interact with each other in various environments to produce a huge range of microbial derivatives and enhance the bio-accessibility and bioavailability of BC. Plant byproducts can be fermented by spontaneous fermentation or by the addition of starter cultures under tailored protocols. In both cases, microbes modify the chemical, biochemical and organoleptic features of substrates, generating edible products that have unique and desirable properties [85]. Lactic acid bacteria, yeasts and molds are commonly involved in most food fermentations (e.g., fruits, vegetables, meats, cheeses, and sourdough baked goods). Among the various technological options, lactic acid and yeast fermentations are considered to be the most natural, sustainable and effective tools for converting plant byproducts into value-added products enriched with high bioavailable BC [85].

Plant byproducts are currently one of the most extensively investigated sources of beneficial health components. The metabolism of phenolic compounds by yeasts and lactic acid bacteria during the fermentation of several plant matrices has been explained [86]. Free or bound phenolic compounds are present in the plant cell wall. If insoluble, they are covalently bound to cell wall structures as hemicelluloses and lignin [87]. Hydrolytic enzymes of yeasts disintegrate the cell wall matrix, leading to the release of bound phenolic compounds (chlorogenic, ferulic, caffeic, and *p*-coumaric acids) and favoring de-polymerization of high molecular weight phenolics [86,87]. Lactic acid fermentation also has a potential role in the hydrolysis of phenolic acid esters (e.g., chlorogenic acid, tannins) by microbial esterases [86]. Moreover, lactic acid bacteria show phenolic acid carboxylase and reductase activity, which are able to metabolize *p*-coumaric, ferulic and caffeic acids into the corresponding reduced or vinyl derivatives, which may exert higher biological activities than the precursors. Specific glycosyl hydrolases of *Lactobacillus plantarum* and *Lactobacillus rhamnosus* convert flavonoid glycosides to the corresponding aglycones, which show high antioxidant and anti-inflammatory capacity [86]. On the whole, plant fermentation increases polyphenol bioavailability with consequent enhancement of in situ radical scavenging potential through the secretion of antioxidant enzymes like superoxide dismutase [85]. Several lactic acid bacteria belonging to *Lactobacillus*, *Lactococcus*, *Leuconostoc* and *Weissella* genera produce exopolysaccharides (EPS), which stimulate the growth of other probiotic bacteria, as well as exerting direct effects on human health, such as immunomodulation effects, antioxidant activity, and cancer prophylaxis. Bioactive peptides are also released during the fermentation process through proteolysis of larger protein molecules. Bioactive peptides exhibit several functionalities in humans, such as the reduction of blood pressure, free radical-scavenging, and antimicrobial activities [85]. Additionally, fermented plant matrices are frequently associated with increased contents of vitamins, short chains fatty acids and insoluble and soluble fibers [88]. The main nutritional and functional effects due to the fermentation process in plant industry byproducts and their applications are summarized in Table 4.

#### 4.2.1. Fermented Fruit Byproducts

Several recent studies have shown fermentation to be an interesting method for valorizing apple pomace and byproducts. For instance, fermentation with autochthonous cider yeasts increased the crude protein, total fat and total dietary fiber content in the apple pomace, with consequent enhancement of nutritional value. Moreover, the content of phenolic compounds, mainly quercetin and phloretin derivatives, as well as that of oleic and linoleic acids, increased after this biotransformation [89]. Fermentation of apple byproducts with a selected binary culture of *Weissella cibaria* and *S. cerevisiae* markedly increased their hydration properties, as well as the total content of soluble and insoluble dietary fibers. In particular, *S. cerevisiae* synthesizes extracellular enzymes that directly react with the plant cell wall, whereas *W. cibaria* synthesizes EPS that might increase the water binding capability of the system. In this context, it is relevant to revalorize apple pomace as a nutritive supplement/ingredient for the production of enriched/fortified foods [89]. Apple byproducts have been successfully recycled through lactic acid fermentation to produce a fermented ingredient suitable to fortify wheat bread [90]. Dough water absorption and bread stability markedly increased without any interference on bread rheology and color. Fortified bread was highly appreciated by sensory analysis. The bread was characterized by a decreased starch hydrolysis index and delayed mold contamination and firming. Using the same approach, date seed flour was fermented with autochthonous lactic acid bacteria and yeasts, and then used as an ingredient for making wheat sourdough bread. The use of this date seed-based ingredient improved the pro-technological (e.g., color, rheology, sensory) and nutritional (e.g., high levels of fiber and polyunsaturated fatty acids) features of bread [101].

Yeast fermentation has the potential to convert pomegranate waste into food supplements with beneficial health effects in vitro—higher than the benefits of the fresh fruit. Solid-state fermentation of pomegranate waste by *A. niger* has been used extensively for the recovery of ellagic acid as a high-value bioactive component. Recently, the use of *S. cerevisiae* as an additional starter to ferment powdered pomegranate husks increased the recovery of ellagic acid 5-fold compared to that obtained with *A. niger* fermentation, and 10-fold compared to the unfermented material. Hydrolytic treatment by *S. cerevisiae* increased the cleavage of punicalagin α and β, which were transformed into punicalin α/β and hexahydroxydiphenic acid at the sugar moiety, which in turn were converted into ellagic acid by spontaneous lactonization [91]. On the other hand, the fermentation of pomegranate juice by *L. plantarum* enhanced the release of ellagic acid. A high human intake of ellagic acid promotes the production of urolithins, which might have critically positive effects on human health, including the inhibition of food-borne pathogens [102].

#### 4.2.2. Fermented Vegetable Byproducts

The manufacture of valuable nutraceutical products through fermentation using microbes and vegetable matrices has been widely encouraged. Tailored microbial fermentations for exploiting the inherent bioactivities of agro-vegetable byproducts seem to be versatile and promising routes. After nine consecutive days of spontaneous fermentation, carrot pulp byproducts are a rich source of organic acids and volatile molecules. During fermentation, the pulp ecosystem is dominated by lactic acid bacteria followed by yeasts that have a great influence on the production of ethanol, acetic and lactic acids. 2-Butanol, as a food additive and flavoring agent, was also found in the fermented carrot pulp. Yeasts are described as 2-butanol-producing microorganisms [92]. Moreover, carrot and tomato byproducts have been used as substrates for lactic acid fermentation using several strains. Fermented byproducts exhibit prominent antimicrobial activity against fourteen pathogenic strains of *Listeria monocytogenes*, *Salmonella* spp., *Escherichia coli*, *Staphylococcus aureus* and *Bacillus cereus*. The antimicrobial effect might be attributed to the ability of lactic acid bacteria to produce antimicrobial compounds such as organic acids (lactic, acetic and propionic acid), diacetyl, and bacteriocins, as well as their ability to change the metabolic profile of phenols, terpenes, aliphatic alcohols, aldehydes, acids and isoflavonoids, which are useful in food preservation. This promising outcome suggests the possibility of applications of fermented byproduct extracts in food production in order to ensure safety and to extend food shelf life [93]. Lactic acid bacteria and yeast fermentations positively affected the antiradical and antimicrobial activities of discarded leaves of red chicory [94]. The high ability to scavenge 2,2-diphenyl-1-picrylhydrazyl free radicals after fermentation was attributed to the release of phenolic compounds, mainly gallic and protocatechuic acids, due to microbial tannase activity. Fermented olive pastes obtained from olive mill waste through spontaneous fermentation occurring in brine with NaCl (6%, *w*/*v*) for 14 days at 20 °C are a further example of a feasible and integrated management strategy for valorizing byproducts. The fermentation of olive mill waste relies on yeasts as the dominant microbial group. Yeasts exhibit metabolic traits thanks to esterase activity and the ability to degrade oleuropein, resulting in the loss of the bitter olive taste and browning capacity [95]. The fermentation of olive mill waste by the sequential inoculum of *S. cerevisiae* and then of *Leuconostoc mesenteroides* resulted in a well-balanced lipid profile, rich in mono- and polyunsaturated fatty acids, and in a very good oxidative stability due to the high concentration of fat-soluble antioxidants. Indeed, the rich composition of fermented olive mill waste suggests its role as a functional ingredient in the food industry [96]. For instance, taralli, a typical bakery product from the South of Italy, was enriched with 20% of fermented olive mill waste to enhance its nutritional and health-promoting properties. The fortified taralli product was a good source of polyphenols, triterpenic acids, tocochromanols, and carotenoids [96].

#### 4.2.3. Fermented Cereal Byproducts

The potential of fermentation to convert cereal-milling byproducts into value-added products has attracted worldwide attention. *L. plantarum* and *W. confusa* have been used to ferment raw and heat-treated maize milling byproduct mixtures of germ and bran [97]. Maize byproduct fermentation resulted in increased free amino acids and peptide concentrations, as well as in phytic acid degradation and enhancement of radical scavenging activity. Moreover, tailored fermentation improved the chemical stabilization of byproducts through the inhibition of the endogenous lipase activity and the prevention of oxidative processes. Fortification of wheat bread with fermented maize byproducts (25% on total weight) increased dietary fiber and protein content compared to the conventional wheat bread. In particular, a relevant increase in protein digestibility (up to 60%) and a significant decrease in the starch hydrolysis index (ca. 13%) were observed after adding fermented maize byproducts to bread, compared to the unfermented ones [97].

Recently, the scientific community has demonstrated great interest in wheat bran fermentation, both when applied alone and in combination with enzymes. Higher nutritional quality was observed after the fermentation of bran with two selected microbial strains of *Lactobacillus brevis* and *Kazachstania exigua* combined with hydrolytic enzymes, mainly xylanase, endoglucanase and β-glucanase, compared with the native one [103]. Bioprocessing caused an extensive breakdown of cell wall structures, entailing an increase in arabinoxylans solubility, more than 11-fold in comparison to that in the native bran [103]. After traditional fermentation of bran, relying on a stable microbiota of lactic acid bacteria and yeasts, a marked increase in soluble dietary fibers (up to 30%) was obtained [104]. Similar results were obtained by fermenting bran with *Lactobacillus bulgaricus* and *Streptococcus thermophylus* combined with commercial bakery yeast [105]. After fermentation, high peptide and free amino acid concentrations (including the functional non-protein γ-aminobutyrric amino acid) were reported, due to the proteolytic activity and to endogenous proteases of lactic acid bacteria [103,104]. The use of *L. brevis* and *Candida humilis* starters after the addition of cell-wall-degrading enzymes positively affected the release and variety of free amino acids (improving protein digestibility) and the release of phenols (mainly ferulic acid) associated with antioxidant properties, inhibition of lipid peroxidation, oxidation of low-density lipoproteins, and anti-inflammatory effects [98]. Spontaneous fermentation carried out by lactic acid bacteriaj, mainly belonging to *Lactobacillus*, *Leuconostoc* and *Pediococcus* genera, and yeasts allowed high release of ferulic acid with an increase of 82% compared to non-fermented bran [104]. Moreover, an improvement in phytase activity occurred when both fermentation and enzymes were used to bioprocess bran, showing high reduction of phytic acid [103,105]. Nevertheless, yeast fermentation strongly increased the folate content of wheat bran by over 40%, and the folate synthesis was also partially ascribed to the presence of indigenous lactic acid bacteria [106]. 

Wheat germ is another high nutritional value byproduct that is discarded during the wheat milling process. Wheat germ is not appreciated for consumption due to some anti-nutritional factors (raffinose, phytic acid, and wheat germ agglutinin) and high lipase and lipoxygenase activities that favor lipid oxidation, and negatively affect the stability of wheat germ [99]. Microbial fermentation of wheat germ is an emerging application that is attracting the interest of scientific and industrial researchers intent on solving this issue. Fermentation of germ wheat with either *L. plantarum* and *Lactobacillus rossiae* significantly decreased aldehydes (usually responsible for the rancidity perception) as well as alcohols, ketones, furanones and lactones (other volatile compounds occurring in lipid oxidation), enhancing the lipase activity and the bioaccessibility of minerals (Ca^2+^, Fe^2+^, K^+^, Mn^2+^, Na^+^, and Zn^2+^). Fermentation also greatly improved the total free amino acid concentration, more specifically lysine, which is the major limiting amino acid of wheat flour. Moreover, it enabled high reduction of raffinose concentration (45%) and production of phenol content (33%), resulting in higher scavenging activity toward free radicals 2,2-diphenyl-1-picrylhydrazyl and 2,2’-azino-bis(3-ethylbenzothiazoline-6-sulfonic acid) [99]. Cytotoxic activity toward cancer cell lines is one of the most promising features of fermented wheat germ. Commercial wheat germ fermented with *S. cerevisiae* showed high anticancer properties in vitro on different human cancer cell lines (including leukemia, melanoma, breast, colon testicular, head and neck, cervical, ovarian, gastric, thyroid, and brain carcinomas), as well as on the prevention of chemical carcinogenesis and some autoimmune conditions [107]. Fermentation (24 h) by selected lactic acid bacteria was able to release quinones (2-methoxy benzoquinone and 2,6-dimethoxybenzoquinone) that were highly present in wheat germ as glycosylated and non-physiologically active forms, thanks to the high β-glucosidase activity. The released form (i.e., non-glycosylated and physiologically active) showed high anticancer features [108].

Recently, selected strains of *L. plantarum* and *L. rossiae* have been used to ferment a milling byproduct mixture of wheat bran and germ [100]. Lactic acid bacteria metabolisms improved the nutritional properties of wheat bran and germ. When fermented, the mixture of wheat byproducts was used as an ingredient for wheat bread making (15% on total weight), resulting in the release of free amino acids and phenolic compounds, which resulted in higher antioxidant activity compared to that of wheat flour bread. Enriched bread was also characterized by high dietary fibers content (6.53%) and low glycaemic index (36.9%), as determined by in vivo tests.

## 5. Conclusions

Fruit, vegetable and cereal byproducts can be considered “treasure troves” of HVC. Nowadays, with serious issues related to the enormous and increasing quantities of FW generated by the industrial processing of these matrices on one side, and the insufficient or not-always-satisfactory strategies for FW disposal on the other side, new solutions for food byproduct management are required. Although a collective commitment and awareness are required to limit this serious issue, and the implementation of legislation for managing the problem appears necessary in order to reduce excessive food loss, the scientific world can significantly contribute to disclosing and setting up new processes aiming to valorize and re-use such byproducts. Against this background, tailored bioprocesses represent promising and effective tools for converting food byproducts from waste to resources, recovering HCV that can be exploited as additional ingredients or supplements in the food, pharmaceutical and cosmetic industries. This review highlighted that tailored enzymatic treatments and fermentations are suitable and natural approaches for recovering bioactive components (often with antioxidant, nutraceuticals and functional properties) from cereal and fruit/vegetable byproducts, which can be reused as food additives in new food, with the final goal to promote beneficial effect for human health. Although enzymatic treatments have already been reported for several cereal and fruit/vegetable byproducts, bioprocesses based on fermentation have only been partially explored, and mainly with respect to apple and pomegranate byproducts. Based on the promising results, a future extension of such natural and sustainable biotechnology to other byproducts is highly desirable in roder to cover the gap.

## Figures and Tables

**Figure 1 molecules-25-02987-f001:**
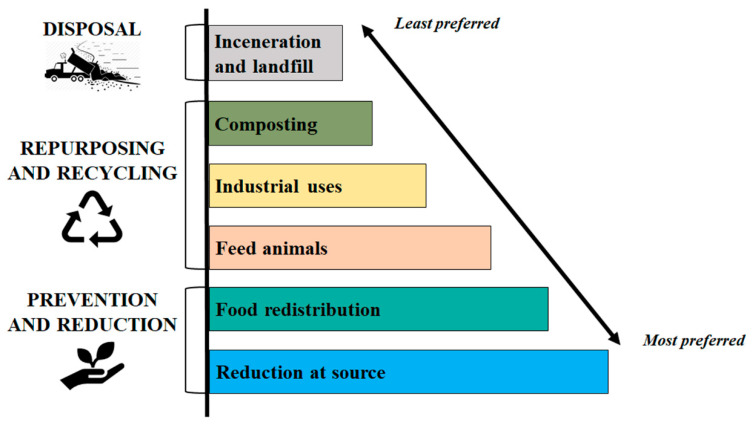
Hierarchy model for the management of food waste and loss [5].

**Figure 2 molecules-25-02987-f002:**
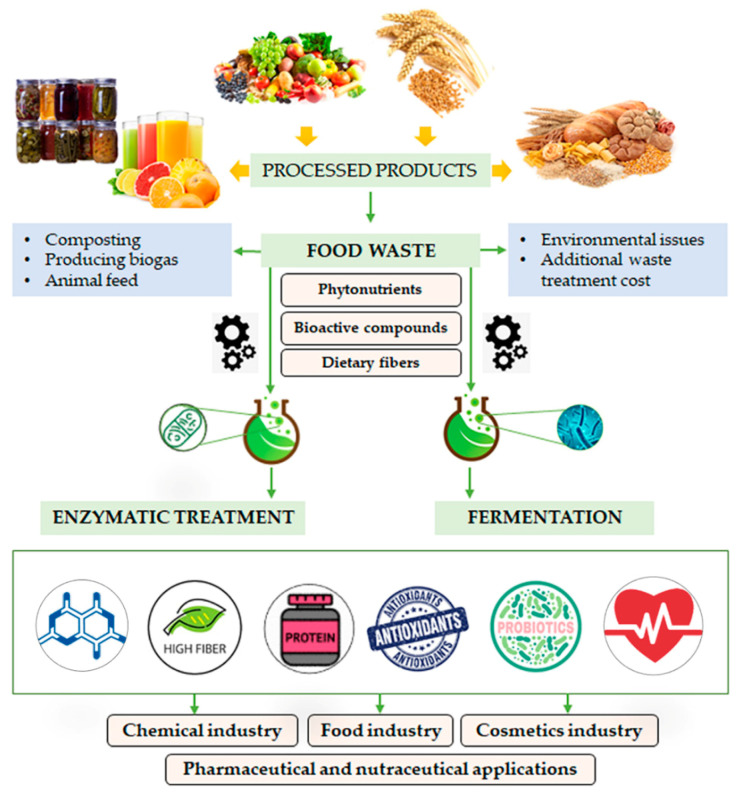
Graphical illustration of the fate of fruit, vegetable and cereal wastes after enzymatic treatment and fermentation.

**Table 1 molecules-25-02987-t001:** Dietary fibers, total phenols and total flavonoids in industrial fruit and vegetable byproducts and their antioxidant activity.

Products	Byproducts	Total Dietary Fibers (%)	Insoluble Dietary Fibers (%)	Soluble Dietary Fibers (%)	Total Phenol (mg GAE 100 g^−1^ DM)	Total Flavonoids (mg QE 100 g^−1^ DM)	Antioxidant Activity			References
							DPPH (mmol TE 100 g^−1^ DM)	ABTS (mmol TE 100 g^−1^ DM)	Others	
**Apple**	Seed	-	-	-	286.1–514.1	-	2.14–4.35	9.15–39.79	-	[11]
	Peel	-	-	-	564	303	22 ± 0.00 ^a^	-	-	[12]
	Pomace	-	-	-	348	-	72.6 ± 1.6 ^a^	84.3 ± 1.6 ^a^	-	[13]
**Avocado**	Seed	-	-	-	57.3	-	41.07 ± 3.58	64.58 ± 1.79	65.69 ± 2.6 ^d^ (FRAP)	[14]
	Peel	-	-	-	63.5	-	31.0 ± 3.69	79.15 ± 3.59	117.51 ± 10.29 ^d^ (FRAP)	[14]
**Banana**	Peel	64.3	56.8	7.4	1500.2	900.3	IC_50_: 55.23 ^b^	-	1.41 ^e^ (Cu^2+^ reduction)	[15]
**Pomegranate**	Peel	-	-	-	45.7	-	92.09 ± 0.36 ^a^	97.14 ± 0.17 ^a^	-	[16]
**Dates**	Seed	9.5	-	-	442.1	-	87.23 ± 1.3 ^a^	-	-	[17]
**Mango**	Seed	-	-	-	ca. 1920	ca. 290	ca. 90 ^c^	ca. 620 ^c^	-	[18]
	Peel	69.8	44.2	24.6	4500	-	-	-	6.96 ^f^	[19]
	Pulp	-	43.1	20.9	100	28.2	-	-	-	[20]
**Citrus**	Seed	-	-	-	7.8	-	IC_50_: 5.53 ± 0.08 ^b^	-	0.92 ± 0.01 ^g^ (AC)	[21]
	Peel	67.4	62.4	4.9	145.5	-	22.4 ± 0.07	-	2.24 ± 0.07 ^g^ (AC)	[22]
**Tomato**	Pomace	-	-	-	86.5	-	-	-	2.46 ± 0.3 ^g^ (AC)	[23]
**Carrot**	Pomace	-	-	-	ca. 500	ca. 320	35 ^c^	-	45 ^c^ (CUPRAC)	[24]

(-), not available. ^a^ Percentage (%) of DPPH or ABTS radical inhibition, ^b^ (µg mL^−1^) at 50% scavenging of DPPH radical, ^c^ mg TE g^−1^ DM, ^d^ mmol Fe^2+^ 100g^−1^ DM, ^e^ µM TR g^−1^ DM, ^f^ µgVit-C Eq. g^−1^ DM, ^g^ mmol TE kg^−1^ DM. DPPH: 2,2-diphenyl-1-picrylhydrazyl. ABTS: 2,2’-azino-bis(3-ethylbenzothiazoline-6-sulfonic acid). FRAP: ferric reducing antioxidant power. TR: trolox equivalent. AC: antioxidant capacity. IC_50_: median inhibition concentration. GAE: gallic acid equivalent. QE: quercetin equivalent. DM: dry matter. *CUPRAC*: CUPric reducing antioxidant capacity assay.

**Table 2 molecules-25-02987-t002:** Overview of the main HVC. Beneficial properties and corresponding examples of cereal byproducts in which they are present and industrial applications of byproducts with other derived technological effects.

Overview of Main HCV			Cereal Byproducts	
Bioactive and Functional Compound	Beneficial Properties	Example of Case Study Demonstrating Beneficial Properties (referred to *)	Byproduct Containing the HCV	Examples of Industrial Application of Byproducts
**Dietary fibers**	Lowering of blood cholesterol*; improvements in large bowel function; reduction of colon tumors; attenuation of post-prandial blood glucose and insulin levels	Animal in vivo study showing a reduction of blood cholesterol levels at the concentration of 0.8 % in standard diet [25]	Wheat bran	Cosmetic and pharmaceutical formulations for moisturization of skin and mucosa; fortification of bread and surimi seafood [26]
**Sterols**	Reduction of blood cholesterol levels*; anti-inflammatory; anti-atherogenic; antioxidant; anti-carcinogenic	Animal in vivo tests showing a reduction of hypercholesterolemia at the concentration of 2455 ± 127 mg 100g^−1^ of total phytosterols [27]		
**Lignans**	Decrease of the production of reactive oxygen species; protection against hormone related breast and prostate cancer; antitumor activity in colon cancer*	In vitro studies showing antitumor activity in colon cancer cells at concentration ≤ 40 µmol L^−1^ [28]		
**Phytic acid**	Antioxidant activity; anticancer activity; reduction of serum lipids and inflammatory processes; prevention of kidney stone formation*; prevention of cardiovascular diseases	Animal in vivo studies showing a reduction of renal calcifications at concentration of 1.08·10^−4^ mol L^−1^ [29]		
**Essential fatty acids**	Antiatherosclerotic effect; antioxidant activity; reduction of blood cholesterol; tryglicerides*	Human in vivo studies showing a decrease of plasma triglycerides in patients treated with 1.0-6.0 g day^−1^ of Omega-3-fatty acids [30]	Wheat germ	Fortification of bread, pasta and biscuits; formulation of nutritional complex for hair loss and health [31]
**Ferulic acid and phenols**	Cardiometabolic protection; anti-diabetic effect; antioxidant activity anti-inflammatory activity*; anti-hypertensive effects	In vitro studies showing the anti-inflammatory effect of hydroxycinnamic acid derivatives at concentration ranging from 1.84 to 378.66 µg g^−1^ [32]	Rice bran	Cosmetical formulations, production of sunscreen and treatment of skin-related disorders [33]
**Tocopherols**	Antioxidant activity*	In vitro studies showing the antioxidant activity of tocopherol at the concentration of 22.5 ± 0.6–170.5 ± 2.0 μg g^−1^ dry weight [34]	Rice germ	Incorporation in infant food [35]
**Anthocyanins**	Colorant effect*	In vitro studies showing colorant effect at concentration between 60 and 120 mg per 100 mL syrup [36]	Corn bran	Fortification of bread, cake and muffin; biosynthesis of natural aroma [37]
**β-glucans**	Anti-cancer property*; immune modulating activity; anti-aging and anti-inflammatory properties	Animal in vivo studies showing anti-tumor effect at both 40 µg and 400 µg dose levels [38]	Brewers´ spent grain	Pasta fortification [39]

**Table 3 molecules-25-02987-t003:** Enzymatic treatments on fruit, vegetable and cereal byproducts for the recovery of main bioactive components.

Byproducts	Enymatic Treatment	Functional Components	References
**Grape skin and seeds**	Pectinase, cellulase and tannase	Proanthocyanidins	[69]
**Grape seeds**	Cellulases and pectinases	Polyphenols	[70]
**Grape pomace**	Tannase Cellulase	Gallic and syringic acids *p*-coumaric acid and malvidin-3-*O*-glucoside	[71]
**Citrus residues**	Pectinase, cellulase, tannase and β-glucosidase	Phenolic	[72]
**Bergamot peel**	Pectinase 62L, pectinase 690L and cellulase CO13 from *Aspergillus* sp. and *Trichoderma* sp.	Pectic oligosaccharides, glycosylated and deglycosylated flavonoids	[73]
**Lime peel**	Laminex C2K from *Penicillium funiculosum*	Pectin	[74]
**Leaves of cauliflower**	Viscozyme L and rapidase	Phenolic compounds and kaempferol-glucosides	[75]
**Tomato residues**	Cellulase and pectinase	Lycopene	[68]
**Apple pomace, passion fruit peels**	Celluclast 1.5L	Pectin	[76,77]
**Fruits, vegetables and grains from cafeteria**	Amyloglucosidase and carbohydrase	Ethanol	[78]
**Rice bran**	Protease type II from *Aspergillus oryzae* and α-amylase type XII-A from *Bacillus licheniformis*	γ-oryzanol, α-tocopherol and other phenolics	[79]
**Rice bran and husk**	Cellulase	γ-oryzanol and vanillic acid	[80]
**Brewer’s grain and wheat bran**	Ultraflo L	Ferulic acid, *p*-coumaric acid, cellulose and arabinoxylan	[81]
**Brewery-spent Grain**	Carbohydrases and peptidase	Carbohydrate- and protein-derived fractions	[82]
**Wheat and rye bran**	Amylase, glucoamylase and xylanase	Ethanol	[83]

**Table 4 molecules-25-02987-t004:** Main nutritional and functional effects of fermentation in plant industry byproducts and related applications.

Plant Matrices Byproduct	Microorganisms Involved in the Fermentation Process	Effects	Applications	Main Outcomes	Reference
Apple pomace	*Saccharomyces cerevisiae*, *Saccharomyces bayanus* and *Hanseniaspora uvarum*	Protein, fat and dietary fiber↑/Phenolic compounds: Quercetin and phloretin derivatives ↑/Oleic and linoleic acid derivatives ↑	-	-	[89]
Apple byproducts	*Weissella cibaria* and *Saccharomyces cerevisiae*	Hydration properties ↑/ Total and insoluble dietary fibers ↑	Fortification wheat bread with 5-10% fermented apple byproducts	Dough water absorption, stability and the content of dietary fibers ↑/Imparted agreeable and specific sensory attributes/Bread hydrolysis index ↓	[90]
Pomegranate waste	*Aspergillus niger* and *Saccharomyces cerevisiae*	Punicalagin isomers and granatin B↓/Ellagic acid ↑	-	-	[91]
Carrot peel	Spontaneous fermentation	Ethanol, acetic acid, lactic acid, 2-Butanol ↑	-	-	[92]
Carrot, tomato and melon byproducts	*Lactobacillus plantarum*, *Lactobacillus casei*, *Lactobacillus paracasei* and *Lactobacillus rhamnosus*	Antimicrobial activity ↑	-	-	[93]
Red chicory leaves	*Saccharomyces cerevisiae*, *Lactobacillus plantarum* and *Lactobacillus hilgardii*	Antimicrobial activity ↑/Antiradical and antioxidant activity ↑/Phenolic compounds: gallic and protocatechuic acids ↑	-	-	[94]
Olive milling waste	Spontaneous fermentation + NaCl (6% *w*/*v*)	Oleuropein concentration ↓/Removing the bitter taste	-	-	[95]
	*Saccharomyces cerevisiae* and *Leuconostoc mesenteroids*	Mono and polyunsaturated fatty acids ↑/Fat-soluble antioxidants ↑	Enriching Taralli, a typical Italian bakery product, with 20% of fermented olive milling waste	Nutritional and health-promoting properties ↑/Polyphenols, triterpenic acids, tocochromanols and carotenoids ↑	[96]
Maize germ and bran	*Lactobacillus plantarum* and *Weissella confusa*	Total free amino acids and peptides ↑/Phytic acid ↓/Radical scavenging activity ↑/Inhibition endogenous lipase activity ↑	Fortification wheat bread with 25% fermented maize byproducts	Dietary fiber and proteins ↑/Protein digestibility ↑/Starch hydrolysis index ↓	[97]
Wheat bran	*Lactobacillus brevis, Candida humilis,* and cell-wall-degrading enzymes	Total free amino acids and protein digestibility ↑/Phenolic compounds and antioxidant properties ↑/Phytase activity ↑	-	-	[98]
Wheat germ	*Lactobacillus plantarum* and *Lactobacillus rossiae*	Volatile compounds occurring in lipid oxidation ↓/Bioaccesibility of Ca^2+^, Fe^2+^, K^+^, Mn^2+^, Na^+^, and Zn^2+^ ↑/Total free amino acids ↑ Raffinose concentration ↓/Phenol content ↑/Scavenging activity toward free radical DPPH and ABTS ↑	Fortification of wheat bread with 4% sourdough fermented wheat germ	Antifungal activity ↑/Shelf life ↑	[58,99]
Wheat germ and bran	*Lactobacillus plantarum* and *Lactobacillus rossiae*	Improvement of functional properties	Fortification wheat bread with 15% fermented wheat byproducts	Total free amino acids and fiber content ↑/Phenolic compounds and antioxidant activity ↑/Glycaemic index ↓	[100]

↑ and ↓ mean “increase” and “decrease”, respectively. DPPH: 2,2-diphenyl-1-picrylhydrazyl; ABTS; 2,2’-azino-bis(3-ethylbenzothiazoline-6-sulfonic acid).

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
