# Peer review of "High-Value Compounds in Fruit, Vegetable and Cereal Byproducts: An Overview of Potential Sustainable Reuse and Exploitation"

_molecules, 2020, doi:10.3390/molecules25132987_

Round 1
Reviewer 1 Report
This review report on the potential of food wastes and by-products from fruits, vegetables, and cereals as a valuable source of high-value bioactive compounds. The manuscript include a detailed overview of the main high-value bioactive compounds and the corresponding sources, as well as a dissertation on the enzymatic and fermentative treatments for their extraction.
The topic is interesting, since in the last years green biotechnologies to revalorize food by-products are highly encouraging to reduce food waste/losses. The manuscript is well-written, figures provide original graphical conceptualization, and tables well-summarized the main recent scientific literatures on the topic. I strongly recommend this review for publication in Molecules.
Only few minor revisions are requested
Table 2. Column “beneficial properties”.. “wheat bran”..”phenolic acids”..Check the numerical order. The column “Area of application” don’t allow an easy understanding and correlation with the corresponding “beneficial properties”.
L.191. Beta-glucans are fibre constituent. Please, clarify the difference between section 2.5 and 2.6
L267 Banana’s
L467. was shown
L483-5. Rephrase the sentence
L494. by-products
L500-504. Please, considers also pulsed electric fields (PEF)
L524. Aspergillus in italics. Please, carefully check throughout the review
L532. enzymes
L537. delete “the”
L555. an useful
L581. Delete “them”
L635. In situ.. should be in italics. Check throughout the manuscript (the same for in vitro)
L637-638. stimulate…exert
Author Response
see in an attached file

Reviewer 2 Report
Sustainable use of food waste discussed in this paper is a timely topic. However, there are several issues to be considered.
- Based on authors’ description, plant includes fruit, vegetable, and cereal but the plant are very broad term. Please define plant in the introduction.
- It seems that food waste from fruits and vegetables are considerable amount throughout the food chain. However, it is believed that food waste from cereal may not be a significant amount. The amount or cost due to food waste from cereal should be mentioned.
- In Table 2, the beneficial properties of bioactive compounds from cereal by-products were summarized. The concentration and study model tested (animal or human) should be added in a separate column.
- The sentences in the section from 2.1. to 2.6 are too general. Please connect the biological function of phytochemicals and dietary fibers to by-products.
- In section 3, the explanation of food waste by-products needs to be more specific.
- In section 3.1., the title should be apple by-products.
- Figure 2 is shown in section 5. It is better to understand if the figure 2 is presented in the introduction.
- There should not be the period at the end of the title of table and in the column of the table.
Author Response
see in an attached file

Round 2
Reviewer 2 Report
The manuscript is improved and good to published. Authors respond well to all comments provided during previous review process.